# Sarconesin II, a New Antimicrobial Peptide Isolated from *Sarconesiopsis magellanica* Excretions and Secretions

**DOI:** 10.3390/molecules24112077

**Published:** 2019-05-31

**Authors:** Andrea Díaz-Roa, Abraham Espinoza-Culupú, Orlando Torres-García, Monamaris M. Borges, Ivan N. Avino, Flávio L. Alves, Antonio Miranda, Manuel A. Patarroyo, Pedro I. da Silva, Felio J. Bello

**Affiliations:** 1Special Laboratory for Applied Toxinology (LETA), Butantan Institute, São Paulo CEP 05503-900, SP, Brazil; andreadiazroa186@gmail.com; 2Institute of Biomedical Sciences, University of São Paulo, São Paulo CEP 05508-900, SP, Brazil; abraham.culupu@butantan.gov.br; 3PhD Program in Biomedical and Biological Sciences, Universidad del Rosario, Bogotá 111221, Colombia; 4Bacteriology Laboratory, Butantan Institute, São Paulo CEP 05503-900, SP, Brazil; monamaris.borges@butantan.gov.br; 5Medicine Faculty, Universidad Antonio Nariño, Bogotá 110231, Colombia; ortorres@uan.edu.co; 6Special Laboratory of Cell Cycle (LECC), Butantan Institute, São Paulo CEP 05503-900, SP, Brazil; ivan.avino@butantan.gov.br; 7Biophysics Department, UNIFESP, São Paulo CEP 04023-062, Brazil; pelopes2@yahoo.com.br (F.L.A.); amiranda@unifesp.br (A.M.); 8Molecular Biology and Immunology Department, Fundación Instituto de Inmunología de Colombia (FIDIC), Bogotá 111321, Colombia; mapatarr.fidic@gmail.com; 9Basic Sciences Department, School of Medicine and Health Sciences, Universidad del Rosario, Bogotá 112111, Colombia; 10Faculty of Agricultural and Livestock Sciences, Veterinary Medicine Programme, Universidad de La Salle, Bogotá 110141, Colombia

**Keywords:** antimicrobial peptide, *Sarconesiopsis magellanica*, *Calliphoridae*, drug, alpha-helix

## Abstract

Antibiotic resistance is at dangerous levels and increasing worldwide. The search for new antimicrobial drugs to counteract this problem is a priority for health institutions and organizations, both globally and in individual countries. *Sarconesiopsis magellanica* blowfly larval excretions and secretions (ES) are an important source for isolating antimicrobial peptides (AMPs). This study aims to identify and characterize a new *S. magellanica* AMP. RP-HPLC was used to fractionate ES, using C18 columns, and their antimicrobial activity was evaluated. The peptide sequence of the fraction collected at 43.7 min was determined by mass spectrometry (MS). Fluorescence and electronic microscopy were used to evaluate the mechanism of action. Toxicity was tested on HeLa cells and human erythrocytes; physicochemical properties were evaluated. The molecule in the ES was characterized as sarconesin II and it showed activity against Gram-negative (*Escherichia coli* MG1655, *Pseudomonas aeruginosa* ATCC 27853, *P. aeruginosa* PA14) and Gram-positive (*Staphylococcus aureus* ATCC 29213, *Micrococcus luteus* A270) bacteria. The lowest minimum inhibitory concentration obtained was 1.9 μM for *M. luteus* A270; the AMP had no toxicity in any cells tested here and its action in bacterial membrane and DNA was confirmed. Sarconesin II was documented as a conserved domain of the ATP synthase protein belonging to the Fli-1 superfamily. The data reported here indicated that peptides could be alternative therapeutic candidates for use in infections against Gram-negative and Gram-positive bacteria and eventually as a new resource of compounds for combating multidrug-resistant bacteria.

## 1. Introduction

Antimicrobial peptides (AMPs) are present in most forms of life, ranging from bacteria to plants, vertebrates and invertebrates [1]. They form part of insects’ complex innate immune system, conferring protection against microbial infections and are synthesized in fat bodies and in hemocytes [2,3,4]. They are released into the hemolymph after their proteolytic maturation to counteract pathogen action, although they can also have local synthesis in several epithelial tissues, such as the gut and epidermis in response to microbes’ exposure in these sites [5,6,7].

AMPs have been identified in necrophagous flies’ salivary glands [8] and are important components of larval excretions and secretions (ES) during this phase of their biological development. Larval external digestion means that digestive enzymes such as serine and metalloproteinases, antibacterial molecules and other biochemicals produced by them are constituents of ES [9,10]. As flies live in an environment contaminated by pathogens’ oral ingestion, their innate defense system is activated, therefore inducing AMPs [11]. These molecules’ mechanism of action (MoA) acting synergistically with other larval ES components, and these living organisms’ mechanical effect (due to their movement in hard-to-heal chronic wounds), thus enables the success of larval therapy. Several works have led to identify, characterize and evaluate the antimicrobial activity of blowfly larvae-derived molecules, including AMPs [11,12,13,14,15,16].

AMPs are small peptides having variable amino acid (aa) composition, usually ranging from 8 to 50 residues in length and their size could be in the order of 2 to 10 kDa. Many of these aas (around 40%) are hydrophobic and have amphipathic properties [17]. They interact with pathogen surface through electrostatic or hydrophobic mechanisms to initiate killing bacteria using mechanisms such as lysis, disrupting microbial homeostasis, membrane permeabilization and rupture, inhibiting protein synthesis or inducing reactive oxygen species (ROS) synthesis causing cell death [18,19,20]. Changes in the primary sequence of the AMPs directly influence their mechanism of action, potency and selectivity against bacteria [21].

Such molecules are usually cationic, having net charges ranging from +2 to +9, with an abundance of lysine and arginine residues [22]. Few anionic antimicrobial peptides (AAMPs) have been recorded from some animal species and human tissues [23]. Insect AMPs are usually small, cationic and have great diversity and repertoire among species [4]. AMPs can be classified according to their structure or function; for instance, there are four structural groups, including α-helical peptides (cecropin and moricin), cysteine-rich peptides (insect defensin and drosomycin), proline-rich peptides (apidaecin, drosocin and lebocin) and glycine-rich proteins (attacin and gloverin) [3,24]. Functional classification tends to be based on target pathogen range instead of any MoA (being very broad in some cases and specific in others) [4,25,26]. Insect-derived AMPs (i.e., *Diptera*, blowflies from the *Calliphoridae* family) have a broad antibacterial, antifungal, antiparasitic and antiviral spectrum, even covering anticancer activities [4,25,26,27].

Pathogen microorganisms’ resistance to the antibiotics currently being used is one of the serious health problems facing humanity. A recent 2018 WHO fact sheet (https://www.who.int/news-room/fact-sheets/detail/antibiotic-resistance) has stated that antibiotic resistance has increased worldwide, reaching dangerous levels. New resistance mechanisms are appearing and spreading throughout the planet day by day, endangering health services’ ability to treat common infectious diseases. Superbugs, such as vancomycin-resistant enterococci, methicillin-resistant *Staphylococcus aureus* (MRSA), carbapenem-resistant *E. coli* and *Klebsiella pneumoniae*, as well as third-generation cephalosporin-resistant strains and drug-resistant tuberculosis are recorded every day in many patients in hospitals worldwide [28]. This situation could lead to the deaths of hundreds of thousands of people who acquire infections around the world every year (i.e., caused by bacteria resistant to one or more current antibiotics), in addition to the loss of hundreds of billions of dollars regarding direct excess healthcare costs and annual productivity losses. 

It has been reported that two million people per year acquire serious infections in the USA due to the action of bacteria which are resistant to current antibiotics, 23,000 of whom die. Excess health service costs could reach up to US$20 billion and annual productivity losses could exceed US$35 billion [4]. The search for new strategies for fighting antibiotic resistance has become a universal priority, one of which could involve insect AMPs and this is why we present a new peptide derived from *Sarconesiopsis magellanica* larval ES.

*S. magellanica* are classified as *Diptera* from the *Calliphoridae* family; they are distributed throughout several South-American countries, such as Argentina [29], Bolivia, Chile, Ecuador, Perú [30] and Colombia. *S. magellanica* can be found in Colombia, even in departments lying at 1200 to 3100 m above sea level (MASL), such as Norte de Santander, Antioquia, Boyacá, Cundinamarca and Bogotá [30,31,32]. This blowfly has been studied by our research group from the point of view of life-cycle, population and reproductive parameters [33], enzymatic characterization [34], larval therapy [35,36] and anti-leishmanial activity [37,38]. It has been demonstrated that their larval ES have potent antibacterial activity [39] and a first AMP called sarconesin has recently been isolated and characterized from these samples. The peptide’s antimicrobial activity against various Gram-positive and Gram-negative bacteria was also evaluated [40].

The main objective of this study was to isolate, characterize and evaluate *S. magellanica* larval ES sarconesin II’s antibacterial action and its MoA for the first time. This was intended to contribute towards the urgent need of developing new alternatives to antibiotics due to the increased incidence of multi-resistant bacteria. The discovery of new AMPs should provide an interesting and promising strategy and could result in a lower probability of the development of resistance observed regarding currently used antibiotics. 

## 2. Results

### 2.1. Purifying S. magellanica ES Sarconesin 

Larval ES crude extract RP-HPLC fractions were lyophilized, suspended in Milli-Q water and tested against *M. luteus*. The sarconesin II fraction having antibacterial activity was eluted at 43.7 min; this peak was collected and chromatographed in the same solvent system onto an analytical C18 column (Figure 1).

The sarconesin II fraction’s antibacterial activity against Gram-positive and Gram-negative bacterial strains was evaluated once it had been purified. Sarconesin II MICs were 15.6 µM for *P. aeruginosa* PA14, 7.8 µM for *E. coli* MG1655 and *P. aeruginosa* ATCC 27853, 3.9 µM for *E. coli* DH5α and *S. aureus* ATCC 29213 and 1.9 µM against *M. luteus* A270 (Table 1). 

### 2.2. Bacterial Growth Curve and Toxicity 

Exponentially grown 10^7^ bacteria/mL were treated with sarconesin II for 7 h, sampling aliquots at different times, and plated on agar. Cell viability was determined by measuring colony forming units (CFU/mL). Sarconesin II MIC proved to be bactericidal against the *E. coli* reference strain after 4 h peptide exposure (Figure 2a). 

Toxicity assays were conducted with 25, 50 and 100 μM sarconesin II concentrations. An MTT proliferation assay was used to determine peptide cytotoxicity, using the HeLa cell line. The compound had more than 95% cell viability at all concentrations (Figure 2b); the amount of formazan produced by living cells did not vary considerably, having no cytotoxic activity, even at the highest concentration tested here. Hemolytic activity was evaluated by determining the amount of human hemoglobin released after incubation with sarconesin II. None of the tested concentrations caused hemolysis (Figure 2c); indicating that sarconesin II had no toxicity. 

### 2.3. Mass Spectrometry and Sarconesin II Characterization

Sarconesin II’s primary structure was obtained by MS/MS; PEAKS software was used to analyze the spectrum and revealed a 13 aa-long molecule, having a 1,439.67 Da mass, with VALTGLTVAEYFR aa sequence (Figure 3). The collision-induced dissociation (CID) spectrum from the mass/charge (*m/z*) of its double-charged ion gave [M + 2H]^2+^, *m/z* 720.3984. 

The ExPASy tool (SIB Bioinformatics Resource Portal) was used to obtain sarconesin II’s physicochemical properties as the peptide sequence was known: i.e., theoretical isoelectric point (pI), molar extinction coefficient (ε) and grand average of hydropathicity (GRAVY) (Table 2). The sarconesin II instability index was calculated to be 2.70, suggesting a stable peptide. The peptide was predicted to have a neutral charge because of having one basic positively-charged Arg (R) residue and one acid negatively-charged Glu (E) aa residue. The considered sequence’s N-terminal was Val (V). The peptide was predicted to have seven out of 13 non-polar hydrophobic aa residues: two Ala (A), Leu (L), Val (V) and one Phe (F). 

Sarconesin II’s other four residues were polar uncharged aa: two Thr (T), one Gly (G) and Tyr (Y). Owing to a V aa residue at the N-terminal, the estimated half-life in vitro suggested that the peptide would remain intact for up to 100 hrs in mammalian reticulocytes, >20 h in yeast (in vivo) and >10 h in *E. coli* (in vivo) [41].

The PEAKS DB database search revealed that the *S. magellanica* ES native peptide fraction might have been derived from the ATP synthase subunit beta, mitochondrial protein previously reported in *Lucilia cuprina*, another *Calliphoridae* blowfly [42]: sequence ID: gi|1321322512, NCBI reference Sequence: XP_023303742.1 and 507 aa total length. Sarconesin II was found to be between residues 260 and 274, covering 3% of the whole protein sequence (Figure 4a). Although the peptide was not submitted to trypsin digestion, the PEAKS software showed (in green) the probable cuts if such enzyme is used. The Arg preceding sarconesin II’s N-terminal Val aa residue and the C-terminal Arg suggested trypsin cuts.

A BLAST search for *Calliphoridae* multiple sequence alignment [43] revealed sarconesin II’s 100% matching identity with the mitochondrial ATP synthase subunit beta (Figure 4b), appearing as a putative conserved domain of the Fli-1 superfamily: flagellar biosynthesis/type III secretory pathway ATPase (Accession: cl25576). Sarconesin II and the conserved domain also appeared in other organisms, such as *Homo sapiens* (Sequence ID: gi|179279|AAA51808.1), *Drosophila melanogaster* (gi|442614522|NP_001259081.1) and *Mus musculus* (gi|31980648|NP_058054.2). Also, a representative model of the ATP Synthase Subunit Beta (Figure 4c), Mitochondrial (PDB ID: 2w6j) was built using Chimera to reveal sarconesin II localization, showing that it is exposed on the surface of the ATPase [44]. 

Sarconesin II was also very similar in length to the ~13–20 aa of other higher vertebrates, such as frogs and rats, and also other arthropod AMPs (Table 3). The peptide also had similar hydrophobicity ~40–69%, having a ~0–3 diverse net charge, like *Plantaricin* with neutral charge [45].

### 2.4. Sarconesin II’s Secondary Structure 

Iterative Threading ASSEmbly Refinement [51] and circular dichroism (CD) were used to predict and study sarconesin II structure. Figure 5a gives the CD spectra, showing a characteristic helix-coil transition spectrum where the peptide had a random conformation in water, characterized by a strong negative band at 200 nm, and becoming more structured as TFE concentration increased. The peptide had a characteristic alpha helix spectrum in 100% TFE concentration, with some distortions which could be attributed to Tyr^11^ aromatic contribution to the spectrum. Aromatic aa provided a positive contribution in 208 and 222 nm bands and a negative contribution in the 195 nm band [52]. 

Sarconesin II aa sequence was entered into I-TASSER providing images of possible structures to further predict sarconesin II’s secondary structure [53], giving a typical alpha-helix structure (Figure 5b). The 13 aas were strongly helical. Furthermore, when bioinformatics analysis was used to arrange the sequence in α-helical wheels, it had no hydrophobic face as reported by the HeliQuest tool; it had a slightly opposite arrangement of hydrophilic and hydrophobic aas, characteristic of an amphipathic α-helical peptide (Figure 5c). 

### 2.5. Mechanism of Action (MoA)

#### 2.5.1. Membrane Integrity

Red fluorescent dye propidium iodide (PI) and carboxyfluorescein diacetate (CFDA) assays were used to study disruption of the bacterial membrane by sarconesin II. PI penetrates damaged cell membranes and intercalates into nucleic acids [55]. PI fluorescent intensity indicates the level of cell membrane integrity as observed in cells without any treatment (PBS) where bacteria did not incorporate PI; treated cells had increased fluorescence when incubated with sarconesin II (Figure 6), suggesting disruption of the cells’ inner membrane. 

An alteration in sarconesin II esterase activity was observed when compared to bacteria control, since CFDA is cell permeant and fluorescent only after exposure to intracellular esterases, thereby confirming membrane alterations.

#### 2.5.2. Sarconesin II Effects on *E. coli* DNA and Protein Profile 

Three sarconesin II concentrations were tested to evaluate whether sarconesin II interacted with bacterial DNA (bDNA) by analyzing the electrophoretic mobility of DNA bands with peptide on an agarose gel (Figure 7a). The bDNA in the gel retardation assay showed that sarconesin II became strongly bound to DNA in vitro because genomic DNA (gDNA) migration from *E. coli* was suppressed. The Figure 7 shows that the peptide bound to the DNA and subtracted charges, which is because it did not migrate, thereby suggesting inhibiting intracellular functions via interference with DNA. 

Cells treated with or without sarconesin II were analyzed in 12% SDS-PAGE to further assess whether sarconesin II affected *E. coli* protein profile. The streptomycin control inhibiting protein synthesis showed no protein profile. Sarconesin II-treated bacteria had the same profile as that for bacteria with just PBS treatment, suggesting no peptide action on proteins (Figure 7b).

#### 2.5.3. Fluorescent Microscopy Assays

Bacterial cytoplasmic membrane integrity was assessed with propidium iodide (PI) staining to investigate whether sarconesin II affected bacterial membrane. The cells exhibited no PI staining in the absence of peptide, indicating that the membranes remained intact. By contrast, cells treated with peptide had intense red fluorescence, showing that sarconesin II could disrupt *E. coli* cell membrane and confirming damaged bacterial membrane permeability (Figure 8).

4′,6-Diamidino-2 phenylindole (DAPI) fluorescent staining was used to evaluate the effect of sarconesin II by confocal microscopy. DAPI intercalates into nucleic acids and yields blue fluorescence when observed in the whole image field. Figure 8 shows that cells treated with ciprofloxacin had less blue fluorescence (as the antibiotic promotes double-stranded DNA breakage). Sarconesin II-treated cells had also partial blue fluorescence thus suggesting *E. coli* DNA fragmentation.

### 2.6. Determining Cell Morphology 

#### 2.6.1. Gram-Stained *E. coli* Cells

*E. coli* culture in logarithmic phase was incubated with or without peptide at 37 °C to observe whether there were any morphological changes. Control PBS and peptide-treated cells were Gram-stained. Microscopic comparison of *E. coli* cell bacterial culture with sarconesin II revealed different morphologies. Figure 9 shows that cells were elongated due to a phenomenon commonly known as filamentation. 

#### 2.6.2. Examining Bacterial Membrane Change by SEM

Scanning electron microscopy (SEM) observations were made to further evaluate and confirm morphological changes detected after treatment with sarconesin II. *E. coli* cells were treated for 12 h with or without sarconesin II MIC. SEM preparations were fixed, dehydrated, coated with gold and examined by microscope, as described in the Materials and Methods section. Figure 10A shows that non-treated *E. coli* cells had intact smooth surfaces, displaying typical morphological characteristics, remaining cylindrically shaped, turgid and smooth, while sarconesin II-treated cells underwent considerable structural changes. After treatment with the peptide, *E. coli* cells appeared as highly elongated, filamentous cells having several holes on their outer membrane (Figure 10B). The surface seemed to have been disrupted; other changes appeared to involve the formation of blebs on cell surface. These results indicated that sarconesin II induced alterations in *E. coli* cell morphology.

## 3. Discussion

Antibiotic resistance constitutes one of the most pressing public health concerns worldwide; multicellular organisms AMPs are considered part of a solution to this problem [57]. This article reports the isolation, characterization and MoA of an efficient AMP purified from *S. magellanica* third-instar larval ES which was named sarconesin II. 

Our RP-HPLC results showed that *S. magellanica* ES could be separated into several fractions. The fraction with a retention time of 43.7 min had antibacterial activity and was named sarconesin II. The *S. magellanica* ES profile looked similar to that previously reported for the same material, having a high peak, followed by sarconesin II sharing the same retention time [40]. Sarconesin II was further purified to confirm homogeneity, using the same chromatographic system (Figure 1).

Sarconesin II had good activity against Gram-negative and Gram-positive bacteria, including *E. coli* K12 MG1655 (Figure 2) [58]. *E. coli* is a Gram-negative bacteria which can use gene mutations and multidrug efflux pumps, resulting in multidrug resistance [59]. Other promising native peptides, such as cecropin A, tenecin 1 and melittin from insects, and magainin II, pexiganan and LL-37 from vertebrates, have also been reported as being able to kill this strain [60]. Other authors have reported AMP activity, having a MIC ranging from 40 to 150 µM, such as maize, lycotoxin I, lycotoxin II and magainin B against *E. coli* DH5α. A 3.9 µM MIC was obtained in this study using the novel sarconesin II; lower concentration was required for killing the bacteria compared to the range of previously reported AMPs [61,62,63]. 

Regarding sarconesin II activity against *P. aeruginosa*, this bacteria has previously been reported as a microorganism developing high resistance to a variety of antibiotics, including aminoglycosides, quinolones and β-lactams [64]. Many AMPs, including GL13K, LL-37, T9W, NLF20, cecropin P1, indolicidin, magainin II, nisin, ranalexin, melittin and defensin, have also had similarly potent antimicrobial effects against *P. aeruginosa* [56,65,66,67,68,69]. The ß-defensin 3 AMP has been shown to have action against the multi-drug-resistant *P. aeruginosa* PA14, having a 32 µM MIC, very close to the MIC found in our study [70].

Sarconesin II has shown antimicrobial activity against Gram-positive bacteria, such as *S. aureus* and *M. luteus*. Previous studies have shown that AMPs have a broad range of antimicrobial activity against bacteria, parasites, fungi and viruses [71,72]. Two alloferons from the blowfly *Calliphora vicina* (the same *Calliphoridae* family as *S. magellanica*) are active against human influenza viruses A and B [73]. Our study did not involve evaluating activity against other microorganisms, but a high probability (0.96) of antiviral activity was found using in silico analysis [74]; this could be interesting for future research, remembering that bactericidal antimicrobial agents are required to treat immunocompromised patients’ infections and other diseases [75]. 

It is known that Gram-negative bacteria are more resistant to antimicrobial agents than Gram-positive bacteria due to their different cellular membranes [76]. The sarconesin II requirement for killing Gram-negative bacteria thus had generally greater MICs than those obtained for Gram-positive bacteria, also demonstrating broad peptide activity killing both. Broad-spectrum AMPs have thus been suggested as an alternative to conventional antibiotics for combatting bacterial infections, as AMPs have rapid killing kinetics, low levels of induced resistance and low host toxicity [68,77,78,79,80]. Time-kill studies [81] for evaluating sarconesin II antimicrobial activity against *E. coli* (Figure 2a) demonstrated that the peptide totally killed the bacterium after 240 min, thereby confirming its bactericidal activity. 

*E. coli* MIC, or the lowest concentration inhibiting visible bacterial growth, was found to be 3.9 µM. We did not evaluate MBC, but a higher drug concentration is probably required to kill bacteria completely [82]. The MIC used in our research for the growth killing assay when plating the bacteria onto agar completely killed the bacteria after 4 h, suggesting that the MIC and the MBC could probably have been the same because sarconesin II caused death during the time evaluated here, also suggesting a bactericidal mode of action.

Faster action has been reported for other *Calliphoridae* family peptides (i.e., *L. sericata* lucifensin) than for sarconesin II. No bacteria-killing assay has been reported for lucifensin, but SEM images after a 60 min treatment showed the effects on bacterial membrane [12]. Just one work was found reporting time-kill curves for *C. vicina* and *L. sericata* total ES against *E. coli*; both *Calliphoridae* had good bactericidal activity against *E. coli*, inhibiting growth over the first four hours [83]. 

Time-kill curves against MRSA ATCC 43300, of an insect defensin-like peptide DLP4 isolated from the black soldier fly *Hermetia illucens* have been reported, showing DLP2 1 × MIC killing effect in 8 h and 4 × MIC in 6 h [84]. This suggested a faster effect for sarconesin II if an increase in MIC against bacteria were to be tested. As predicted in silico, the time that sarconesin II could remain stable without degradation in *E. coli* was 10 h [41]; this could be enough time for complete sarconesin II action. It has been reported that some native drugs could gradually lose their effect after 24 h, and bacteria escaping drug action can multiply at a faster rate in suitable conditions [82]. Sarconesin II action is faster than its degradation rate. It is known that the biological ability of AMPs in serum may become largely reduced, thereby highlighting the importance of their transportation and storage [85]. Sarconesin II’s instability index [86] suggested a stable peptide, an important role regarding any biological drug’s application. 

Encouragingly, concentrations where sarconesin II was active did not have any toxicity or hemolytic activity regarding either HeLa or erythrocyte cells, even when tested at concentrations 100 times higher than bacterial MIC (Figure 3), suggesting the compound’s high selectivity. Nagarajan obtained similar results when testing the NN2_0018 peptide which also has antibacterial activity against resistant bacteria [58]. Recently discovered peptide LGH2 had low toxicity when evaluated against sheep RBC at 4 μM [87]. When evaluated at a concentration similar to that for sarconesin II (100 µM) it had 100% hemolysis, while that for sarconesin II continued without having toxicity. Accordingly, sarconesin II did not have any toxicity but did have potent antibacterial activity, making it a promising candidate for use in therapeutics.

The sarconesin II primary sequence obtained by HPLC-ESI-Orbitrap-MS had the same identity as that for the ATP synthase beta subunit protein conserved domain. Sarconesin II was also detected in the organisms of other larval extracts, as the BLAST search showed, and has been previously identified in research into human breast cancer MCF-7 and MDA-MB-231 cell lines, and as proteases in basidiomycete *Amanita virgineoides* [88,89]. Sarconesin II is a derivative of the subunit beta ATP synthase reported in a broad range of organisms. It is 13 aa long, making it a shorter sequence having ideal features for clinical use [63]. Sarconesin II probably did not have any toxicity because it is part of an ATP protein conserved domain which is present naturally in human cells and in all eukaryotic cells [88].

Sarconesin II was not subjected to tryptic cleavage but had particular trypsin cuts which may have involved proteolysis of the ATP synthase beta subunit protein having a 54.6 kDa molecular weight (https://www.bioinformatics.org/sms/index.html); the presence of a protein having this molecular weight was confirmed using the *S. magellanica* ES protein profile [34]. This suggested that the protein could have been in previous contact with the enzyme, making sarconesin II a sub product of the protein. The 3D representative model also highlighted sarconesin II as being exposed on ATPase surface, facilitating the cutting. This has also been reported for sarconesin, knowing that ES contain chymotrypsin and other enzymes [40]. This suggests that proteins may play a role in blowflies’ innate immunity during extracellular digestion, helping as a substrate for creating AMPs, probably because insect AMPs are secreted by fat body cells into the hemolymph (its action as humoral immunity factors has already been described) [90]. 

APD analysis revealed sarconesin II-similar AMPs and showed that all of them had antimicrobial activity against both Gram-negative and Gram-positive bacteria, having similar or higher MICs than sarconesin II. Moreover, VmCT1 (an α-helical AMP from scorpions) has been reported to have activity against cancer cells [50,91] and temporin-HN1 has been reported to have antifungal activity [46]. Our study found no sarconesin II toxicity against a breast cancer cell line; however, it could be interesting to test it against other cell lines and evaluate its antifungal activity. 

APD gave sarconesin II 38.8% similarity with the H4-(86–100) peptide. This can be considered a good similarity percentage as the APD database contains 3,055 peptides. Similar percentages regarding other AMPs were found at the APD2 database, i.e., H4-(86–100) was reported as having 35.3% similarity with Temporin-LTb [45]. It should be stressed that sarconesin II had action in DNA in our study like the H4-(86-100) peptide which has previously been reported to cause an inhibitory effect on DNA gyrase [47]. 

The CD spectra indicated that sarconesin II had a random coil in water and adopted an α-helical structure in 70% TFE (Figure 5a). This pattern indicated that sarconesin II was prone to assuming a specific conformation when interacting with membrane-mimicking agents like TFE or SDS. Alpha-helical characteristics are stronger in TFE. This agreed with TFE’s known properties for promoting helical structures in peptides [92,93]. Peptide secondary structure was observed by I-TASSER and it has been reported that increased helix propensity also increases antimicrobial potency [94]. AMPs having α-helices are magainin, cecropin and cathelicidin; however, they can perform their functions through interactions with intracellular targets or by disturbing cell processes, as well as inhibiting cell wall, nucleic acid or protein synthesis [95].

An α-helical structure has been found for other members of the Fli family [86]. The sarconesin II sequence was arranged in α-helical wheels by bioinformatics analysis to further characterize its structural properties, revealing a slightly opposite arrangement of hydrophobic aa [96]; this is characteristic of amphipathic α-helical peptides, and is also known to be important in amphipathic arrangement [97]. The ability to structure an α-helix, without this arrangement is not sufficient to provide potent, broad-ranging, antimicrobial activity, as demonstrated by scrambled peptide P19(6|s) [98,99]. The importance of amphiphilic helices has been discussed regarding serval AMPs, and it has previously been shown for another *Calliphoridae, Sarcophaga peregrina* (flesh fly), whose sarcotoxin IA and Pd peptides consist of two amphiphilic helices, having hydrophilic and hydrophobic faces [100,101,102]. 

Moreover, such hydrophobic arrangement can facilitate recognition of hydrophobic lipids on bacterial membranes. Sarconesin II has a C-terminal positively charged Arg, frequently occurring in protein active or binding sites, giving a higher probability of interaction with negatively-charged non-protein atoms [103]. Interestingly, some peptides have been classified as being glycine-rich; we cannot affirm that sarconesin II is part of this family, considering that it just has 13 aa, where Gly represents 7% of it. Glycine-rich peptides come mostly from insects and are active against Gram-negative bacteria [24,104]. 

A mechanism for killing bacteria involving membrane disruption is probably particular for hydrophobic, moderate to strong amphipathic, α-helical peptides, involving Arg, as found for sarconesin II and previously reported for AA230 arenicin-3 as an amphipathic molecule rich in Arg and hydrophobic aas [105]. These characteristics have also been found for sapecin which has essential residues for membrane channel formation in its sequence, such as Asp and Arg. The opposite is true for plectasin where its sequence does not even have most of the hydrophobic residues at a specific site in the a-helix chain [106].

Increased sarconesin II in esterase assays indicated bacterial membrane alteration; it has also been reported that esterase activity in *E. coli* seems therefore to function as a stress indicator rather than a viability parameter [107]. After PI staining, sarconesin II-treated *E. coli* cells had increased fluorescence (Figure 6 and Figure 8) since PI cannot diffuse into viable cells or dead cells having an intact membrane. PI confirmed sarconesin II entering into *E. coli* cells and binding to DNA, thus confirming its action on the membrane. Knowing that PI stains dead cells as a result of porous membrane, other peptides have been reported such as NK-18, thereby showing the alteration of both; the outer and inner membrane were disturbed after treatment [108], as found for sarconesin II. The AMP cecropin has also been reported to induce inner membrane perturbation shown by high PI incorporation [109,110]. Bacteria’s killing effect depends on membrane damage and inhibiting the membrane’s functional proteins which can be critical for bacteria, even if there is no complete lysis [111].

This could imply that the peptides were not only associated with the plasma membrane but could also enter cells. This prompted us to look for some intracellular-targeting MoA for this peptide [112]. Peptides that affect Gram-positive bacteria usually act on the membrane; however, no parameters enable discriminating between peptides acting on Gram-negative or Gram-positive bacteria. Malanovic et al., [113] reported no apparent preference for targeting only Gram-positive or Gram-negative bacteria regarding hydrophobicity and aa residue charge and that AMP secondary structure could be identified when examining the APD [114].

One bactericidal drug mechanism concerns the induction of DNA and protein damage [115]. No action was observed in proteins; sarconesin II DNA damage was perhaps caused by fragmentation, as shown by DAPI assay fluorescence [116,117] or by sarconesin II interaction with and binding to bDNA during gel retardation assay. One of the better-studied peptides interacting with DNA is buforin II [118]. Rondonin is another neutral peptide having a similar MoA regarding DNA as that for sarconesin II [119]. Other authors have reported that AMPs such as insect defensins DPL2 and DLP4 have differential ability against different species (as evidenced on agarose gel) regarding their binding to DNA, having variable binding for *E. coli* CVCC1515 or *S. typhimurium* ATCC 14028 [84]. Tryptophan in the sarconesin II sequence could also explain the double MoA on membrane and DNA as this aa was shown to be involved in hydrophobic interactions with bacterial membranes and DNA [62]. 

Both MoA discovered for sarconesin II are probably important for increasing antibacterial activity against Gram-negative and Gram-positive bacteria; this was also observed for NK-18 [63]. Another example of double action is indolicidin which can kill bacteria, fungi and HIV. It has antifungal activity by causing cell membrane damage and also kills *E. coli* by penetrating cells and inhibiting DNA synthesis [111].

Morphological changes to the bacteria detected by SEM provide evidence that sarconesin II had an effect on *E. coli* cell membrane; a similar effect was obtained for NK-18 [63]. Its MoA has been related to cell membrane permeabilization. The Gram and SEM images revealed bacterial elongation; this morphology was also found for indolicidin which inhibits DNA synthesis, leading to *E. coli* filamentation [120]. This was indicative of cell division inhibition which correlated with a lack of thymidine incorporation into cells, suggesting that it was interacting with DNA [121]. Once the peptide was internalized, it led to the formation of membrane vesicles [111]. It is worth noting that the SEM images did not show complete membrane rupture and, as sarconesin II is an α-helical peptide, it may have created pores on the membrane [111,122].

Impaired bacterial cell division also depended on intracellular elastase-like serine protease activity which can proteolytically activate cathelicidins. Sarconesin II induced filamentation, resulting in the formation of filamentous bacteria having arrested septation [123]. Such morphology indicated a bacterial stress response and has been observed in bacteria responding to damage from low doses of antibiotics, starvation and ROS and nitrogen species [124,125]. Furthermore, the appearance of filamentous cells first suggested an induced SOS repair system response via DNA damage, also called SulA-dependent filamentation [126,127].

A second veto pathway responsive to DNA damage has also been identified in *E. coli*, known as the sfi-independent pathway causing filamentation in sfi cells via induction of the LexA regulon. [128]. Hill et al., have reported that LexA is necessary and also arrests DNA replication for inhibiting cell division. Sarconesin II might thus have induced filamentation in *E. coli* by inhibiting DNA synthesis [129].

Some AMPs have been reported as being membrane disruptive, such as alamethicin, magainin 2, cecropin and nisin, while some are membrane non-disruptive but do have internal action on DNA, RNA or proteins, i.e., buforin II, pleurocidin and PR39 [121]. Sarconesin II could thus be a membrane non-disruptive peptide, also having internal action on DNA.

This paper has thus provided a certain understanding of the whole sarconesin II antibacterial MoA. It could reasonably be assumed that sarconesin II’s neutral, hydrophilic characteristics first initiate an electrostatic interaction with the bacterial cell membrane’s negatively charged components. The peptide then induces outer and inner membrane permeabilization and depolarization, creating pores which the peptide can then use to enter cells. The budding or “wart”-formation can also lead to cell envelope destabilization, as seen by Gutsmannn [130]. Along with peptide internalization, sarconesin II could interact with DNA through electrostatic interaction, as found by Yang et al. [63]. It would then spontaneously bind to DNA, causing filamentation in bacterial cells, inhibit the cell repair function and lead to killing bacteria. As sarconesin II has two MoA it can easily kill Gram-positive and Gram-negative bacteria and is a potent AMP, that might become a novel tool for combating resistant bacteria. 

## 4. Materials and Methods 

### 4.1. Fly Source and S. magellanica ES Collection

*S. magellanica* were collected from a previously established colony at the Universidad del Rosario’s Medical and Forensic Entomology laboratory. All individuals were maintained at 20–25 °C, with 60–70% relative humidity and a 12/12 h photoperiod, being fed daily liver and sugar solution supplement to continue the biological cycle. Around two thousand Instar III larvae were incubated with the selected bacteria for the immune challenge; the ES were obtained after larval disinfection, as described previously [40].

### 4.2. Bacterial Strains

The strains used were the multi-drug resistant *Pseudomonas aeruginosa* PA14, *P. aeruginosa* 27853, *Escherichia coli* MG1655, *E. coli* DH5α, *Staphylococcus aureus* ATCC 29213 and *Micrococcus luteus* A270; they were obtained from the Butantan Institute’s Special Laboratory for Applied Toxinology (LETA) (São Paulo, Brazil), while the resistant *P. aeruginosa* PA14 strain was kindly donated by Dr. Beny Spira (USP, Brazil). 

### 4.3. Antimicrobial Assays 

Minimum inhibitory concentration (MIC) was assayed according to the standard method [131]. Exponential growth phase cultures were diluted in poor nutrient broth (PB) to 5 × 10^4^ CFU/mL (DO = 0.001) final concentration [11,132,133]. The fractions’ antimicrobial effects were evaluated by liquid growth inhibition assay, using 96-well sterile plates; 20 µL serial dilutions of the fractions were incubated with bacteria at a final 100 µL volume. Sterile water, PB and streptomycin were used as growth and growth inhibition controls. The plates were incubated for 18 h at 30 °C. Absorbance was measured at 595 nm [119].

The time-kill studies involved using a final inoculum of around 1.5 × 10^7^ CFU/mL in a final 2 mL volume in a polypropylene tube. The sample and the control were incubated at 37 °C. Serial 100-fold dilutions were prepared in distilled water at each sampling time (0, 30, 120, 240, 360 and 420 min), when necessary. A 10 µL aliquot of the diluted and/or undiluted sample plated in triplicate on LB agar were incubated for 24 h at 37 °C; the colonies were counted. Activity was considered to be bactericidal when the original inoculum was reduced by ≥3 log CFU/mL (99.9%); bacteriostatic activity was defined as a reduction in the original inoculum by <3 log CFU/mL [105]. 

### 4.4. Acid and Solid Phase Extraction 

The ES were incubated with 2M acetic acid for five minutes and centrifuged for 30 min at 4 °C. The supernatant was eluted using Sep-Pack C18 cartridges (Waters Associates, Milford, MA, USA), using two acetonitrile (ACN) concentrations in water (0% and 80%). The hydrophobic part (80%) was lyophilized and reconstituted in 2 mL trifluoroacetic acid (0.05% TFA).

### 4.5. Peptide Purification

Reverse-phase high-performance liquid chromatography (RP-HPLC) was used to fractionate the hydrophobic part of the ES, using a semi-preparative C18 Jupiter column (Phenomenex International, Torrance, CA, USA 10 µm; 300 A; 10 mm × 250 mm), with an elution gradient from 0 to 80% in 60 min, at 2 mL/min flow rate. The fractions were manually collected and absorbance was monitored at 225 nm [133]. The fractions’ antibacterial activity was determined, and fractions eluted with 40% ACN were fractionated on an analytical Jupiter C18 column (10 µm; 300A; 4.6 mm × 250 mm), at ACN concentration ranging from 30% to 45% [134]. The purified fraction’s (sarconesin II) antibacterial activity was evaluated against that of previously reported strains. 

### 4.6. Toxicity 

Cytotoxicity (CC) activity was evaluated against HeLa (human cervical carcinoma) cells kept in DMEM culture medium, supplemented with 10% heat-inactivated bovine serum and antibiotic–antimycotic solution (100 units/mL penicillin, 100 g/mL streptomycin and 25 g amphotericin B) in 5% CO_2_ at 37 °C [135]. MTT assays were used to evaluate CC. 5 × 10^5^ cell/well were seeded in 96-well plate for 24 h and eight, two-fold serial dilutions of sarconesin II (starting at 100 µM) were allowed to react with the cells for 24 h. 30% DMSO and medium were used as control [9]. 5 mg/mL of MTT reagent were incubated for 4 h at 37 °C and dissolved with 150 µL isopropanol. Absorbance was measured at 550 nm and CC was determined, using the formula CC% = (peptide treated cells/peptide untreated cells) × 100 [40,136].

Hemolytic activity was assessed using human erythrocytes. Cells were collected in 0.15 M citrate buffer, at pH7.4 and washed three times with PBS to a final 4% (*V/V*) concentration. Eight two-fold serial dilutions of sarconesin II were evaluated at 100 µM final concentration and incubated for 1 h at 37 °C. Hemolytic activity was determined by measuring absorbance (Abs) at 405 nm and calculated as a percentage of 100% lysis control (0.1% Triton X-100); % hemolysis = (Abs sample–negative Abs)/(positive Abs − negative Abs) [137].

### 4.7. Mass Spectrometry and Sarconesin II Identification

Mass spectrometry analysis involved using a LC-MS/MS on an LTQ-Orbitrap Velos (Thermo Fisher Scientific, Waltham, MA, USA) coupled to an Easy-nLCII liquid nano-chromatography system (Thermo Scientific). The chromatographic step involved using 5 µL of each sample automatically on a C18 pre-column (100 µm I.D. × 50 mm; Jupiter 10 µm, Phenomenex Inc., Torrance, CA, USA) coupled to a C18 analytical column (75 µm I.D. × 100 mm; ACQUA 5 µm, Phenomenex Inc.). Samples had been previously concentrated in a vacuum centrifuge and diluted in 15 μL 0.1% formic acid (FA). The eluate was electro-sprayed at 2 kV and 200 °C in positive ion mode. Mass spectra were acquired by FTMS analyzer; full scan (MS1) involved using 200–2000 *m/z* (60,000 resolution at 400 *m/z*) as mass scan interval with the instrument operated in data dependent acquisition mode. The five most intense ions per scan were selected for fragmentation by collision induced dissociation. The minimum threshold for selecting an ion for a fragmentation event (MS2) was set to 5000 cps [40].

MS/MS peak list files were screened against a bank database constructed with the *Lucilia* proteins obtained from UNIPROT and NCBI [138,139] to determine their aa sequence. They were compared using PEAKS 8.5 (Bioinformatics Solutions Inc., Waterloo, ON, Canada) search software, using the following parameters: oxidation considered as variable modification, 10 ppm precursor mass tolerance and 0.6 fragment ion mass tolerance [137].

### 4.8. Bioinformatics Tools

The sarconesin II sequence was searched for similarity against *Calliphoridae* proteins registered in the National Center for Biotechnology [138] public database, using the Basic Local Alignment Search Tool (BLASTp) [43], with default parameters [140]. 

The antimicrobial peptide database APD [45] and the ClassAMP prediction tool were used to classify AMPs [74].

Sequences’ physical-chemical parameters were calculated using the Swiss Institute of Bioinformatics (SIB) website’s ExPASy bioinformatics resource portal’s ProtParam tool [41]. The online I-TASSER server available on the Yang Zhang laboratory website [51] was used to obtain a three-dimensional (3D) image of peptide secondary structure [53]. 

The Chimera structure prediction tool (accessed through the European Bioinformatics Institute) was used to obtain the location of sarconesin II in the representative model of ATP Synthase Subunit Beta, Mitochondrial (PDB ID: 2w6j) [44]. The molecular weight of the protein was calculated at the following link, https://www.bioinformatics.org/sms/index.html.

### 4.9. Circular Dichroism (CD)

The peptide’s far-UV (190–250 nm) CD spectrum was recorded on a Jasco J 810 spectropolarimeter (Jasco Inc., Tokyo, Japan) at 25 °C, using a 0.1 cm path length quartz cell. CD spectra were recorded after accumulating 4 runs. The peptide was analyzed in pure water and 30, 70 and 100% *v/v* solutions of 2,2,2 trifluoroethanol (TFE) in water. Fast Fourier Transform (FTF) was used to minimize background effects [141].

### 4.10. Mechanism of Action (MoA)

#### 4.10.1. Membrane Damage and Esterase Activity 

The *E. coli* suspension (2 × 10^8^ CFU/mL) was incubated with or without sarconesin II MIC solution at 37 °C for 4 h. The cells were washed with PBS three times by spinning (2000 rpm, 5 min) and bacterial membrane integrity was assessed by fluorometry and microscopy using propidium iodide (PI 60 µM) for 15 min and measured with 485/620 nm excitation/emission wavelengths. PI microscope slides were prepared by placing 10 µL of the mixture, covered with a glass coverslip, and observing them by microscope [55,142]. Microscopy was performed using a TCS SP8 confocal laser scanning microscope (Leica, Mannheim, Germany); Leica Application Suite X (LAS X) software was used to process the images. Esterase activity was evaluated by incubating in the dark, washing cells with 250 µM 5(6)-carboxyfluorescein diacetate (CFDA) for 30 min, followed by measuring fluorescence at 485/535 nm excitation/emission wavelengths [55,143].

#### 4.10.2. DNA Binding Activity and Fluorescence Microscopy

A gel retardation assay determined whether sarconesin II had DNA-binding activity. *E. coli* DNA was purified using the method described by Landry et al., (1993) [144]; 500 ng gDNA were incubated for an hour with 25, 50 and 100 µM sarconesin II. The mixtures were subjected to gel electrophoresis on a 0.8% agarose gel [145]. The previously washed peptide and bacterial cell mixtures were fixed on slides, permeabilized with ethanol, and stained with 4′,6-diamidino 2-phenylindole (DAPI) for visualizing the DNA using a confocal microscope. 

#### 4.10.3. Total Protein Profiling of *E. coli* Cells Treated with Sarconesin II

*E. coli* cells were grown in Luria-Bertani (LB) medium and cell suspension turbidity was adjusted to a final 1 × 10^8^ CFU/mL concentration, measuring bacterial suspension optical density (OD) at 595 nm (OD595). After growth, cells were harvested by spinning at 4000 g for 10 min at 4 °C. The pellets were washed twice with PBS and 10^8^ cells were suspended in 1 mL PB medium for further treatment at different sarconesin II concentrations (25, 50 and 100 µM) for 12 h at 37 °C. 

The samples were spun at 13,000 rpm for 3 min at 4 °C; supernatants were discarded. This was followed by adding 225 µL lysis buffer (25 mM Tris–HCl, pH 7.5, 100 mM NaCl, 2.5 mM EDTA, 20 mM NaF, 1 mM Na_3_VO_4_, 10 mM sodium pyrophosphate, 0.5% Triton X-100, protease inhibitor cocktail) containing 30 mM IAM to the pellets and sonication was performed in 3 bouts of 30 sec each on ice. The supernatants were collected by spinning at 14,000 rpm for 15 min at 4 °C and stored at −20 °C. Supernatant protein concentration was determined based on absorbance at 280 nm using a NanoDrop 2000 spectrophotometer (Thermo Fisher Scientific, Waltham, MA, USA). 

*E. coli* cells total protein profiling was carried out using 12% sodium dodecyl sulphate-polyacrylamide gel electrophoresis (SDS-PAGE). The samples were incubated with SDS-loading buffer at 90 °C for 3 min before electrophoresis and then 25 µg of each sample was applied directly to the polyacrylamide gel. The total running time was 3 h at 120 V. The gel was then stained with silver nitrate or Coomassie Brilliant Blue R-250. Bacterial cells were incubated with streptomycin (a protein synthesis inhibitor) as positive control and incubated without antibacterial agents as negative control [146].

### 4.11. Determining Cell Morphology

#### 4.11.1. Gram Assay

*E. coli* culture in logarithmic phase of growth was diluted to ~0.04 OD 600 and incubated for 4 h with sarconesin II at 37 °C. Control PBS and peptide-treated cells were Gram-stained. Images were acquired using an IX81 microscope (Olympus, Tokyo, Japan), 100 × 1.35NA lens with Cell R software [147]. 

#### 4.11.2. Scanning Electron Microscopy (SEM)

Mid-log phase *E. coli* cells (1 × 10^8^ CFU/mL) were incubated with MIC sarconesin II for 12 h at 37 °C. The bacterial cells were then spun and washed three times with 0.1 M PBS (pH 7.2) and fixed overnight at 4 °C with 2.5% glutaraldehyde. After washing twice with PBS, the cells were post-fixed on cover glasses with 1% osmium tetroxide (OsO_4_) in 0.2 M sodium cacodylate buffer for 1 h, dehydrated in a graded ethanol series (30%, 50%, 70%, 95% and 100%) for 15 min each time, and dried by the critical point method drying from liquid CO_2_. Gold-palladium was sputtered on samples and observed on a QUANTA 250 SEM (FEI, Hillsboro, OR, USA) at 12.5 kV [148]. 

#### 4.11.3. Statistical Analysis

GraphPad Prism software (version 7.00, La Jolla California USA) was used for all statistical analysis. One-way ANOVA was used for statistical comparison of combination treatment regarding toxicity assays, using (α = 0.05) with Dunnett’s multiple comparisons test. Data is presented as mean ± standard deviation (SD).

## 5. Conclusions

The present study described the process of obtaining, characterizing and evaluating the antibacterial activity of a new AMP derived from the larval ES of *S. magellanica*, named sarconesin II. This AMP was obtained and purified using RP-HPLC. The evaluation of the sarconesin II fraction’s antibacterial activity against Gram-positive and Gram-negative bacteria was demonstrated by MIC and measuring CFU. The peptide had no cytotoxicity in the tests used here. Some of this AMP’s relevant physicochemical characteristics obtained by MS/MS and spectrum analysis were: having a 13 aa sequence (VALTGLTVAEYFR), seven non-polar hydrophobic aa residues and another four polar uncharged aa (established predictively). It had a neutral charge because of having one basic positively-charged Arg (R) residue and one acid negatively-charged Glu (E) residue. The PEAKS database search revealed that the native peptide fraction might have been derived from the ATP synthase β subunit, a mitochondrial protein previously reported in *Lucilia cuprina*. A BLAST search for *Calliphoridae* multiple sequence alignment revealed sarconesin II’s 100% matching identity with the mitochondrial ATP synthase β subunit. This peptide’s secondary structure had a characteristic α-helix, predicted by I-TASSER and CD. The tests used to determine sarconesin II MoA on bacteria recorded disruption of the cells’ inner membrane (PI), accompanied by alterations in sarconesin II’s esterase activity, thereby confirming membrane alterations (CFDA), intracellular function inhibition via interference with DNA (DNA band electrophoretic mobility), without affecting the protein profile (SDS-PAGE) and damage to bacterial DNA (fluorescence microscopy). The peptide’s effect on bacterial cell morphology (Gram staining) revealed elongation, a phenomenon commonly known as filamentation, while using SEM demonstrated sarconesin II action on bacteria (having a highly elongated appearance) as filamentous cells having blebbing on their outer membrane. This AMP represents a new weapon for fighting against pathogenic microorganisms, acting mainly on both Gram-positive and Gram-negative bacteria, and showing so far good efficacy against an antibiotic-resistant pathogen such as *P. aeruginosa* PA14.

## Figures and Tables

**Figure 1 molecules-24-02077-f001:**
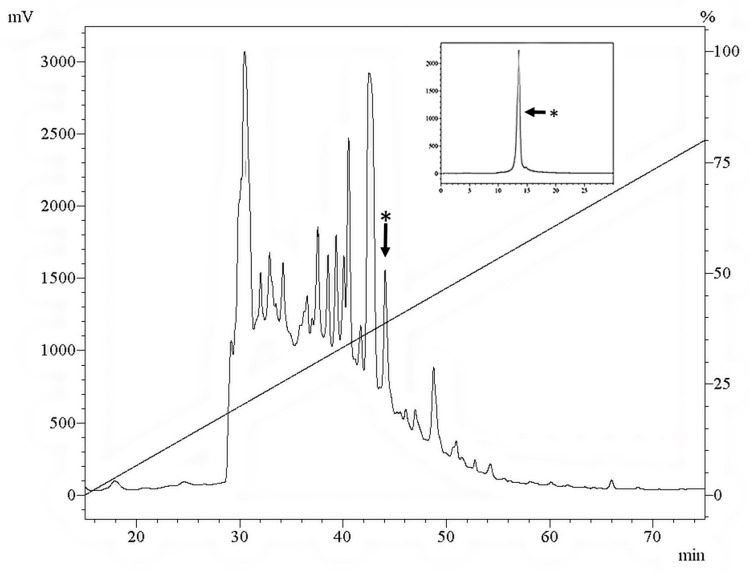
RP-HPLC profile of lyophilized larval ES crude extract at 225 nm eluted on a Jupiter C18 Column (10 mm × 250 mm, 10 µm; 300 Å), using a 0–80% B linear gradient in 80 min, solvent A: 0.05% TFA/ACN and solvent B: 0.05% TFA/H_2_O at 1.5 mL/min flow rate. Inset: the sarconesin II fraction labelled with an asterisk was chromatographed in the same solvent system on a Jupiter C18 column (4.6 mm × 250 mm, 10 µm; 300 Å) using a 30–45% B linear gradient in 60 min.

**Figure 2 molecules-24-02077-f002:**
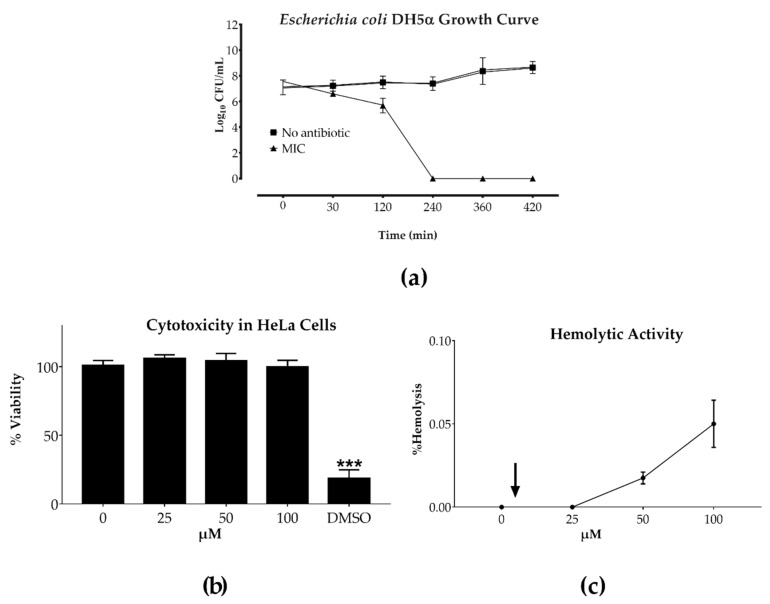
Sarconesin II’s effect on toxicity assays. (**a**) Growth curve for *E. coli* DH5α incubated with sarconesin II MIC. Bacterial growth was inhibited at 240 min. (**b**) HeLa cell cytotoxicity measured by MTT tetrazolium assay. Cells were incubated with sarconesin II at 25, 50 and 100 μM for 24 h. Untreated HeLa cells were used as negative control and HeLa cells treated with 30% DMSO were used as positive control. (**c**) Hemolytic activity against human red blood cells (RBC). Sarconesin II was tested at 25, 50 and 100 μM concentrations. PBS was added without peptide for determining 0% hemolysis. The arrow indicates the concentration at which the fraction had antimicrobial activity. The average of each experiment done in triplicate is presented in individual columns as mean ± SD. One-way ANOVA followed by post hoc Dunnett’s multiple comparison test was used. *** *p* < 0.001.

**Figure 3 molecules-24-02077-f003:**
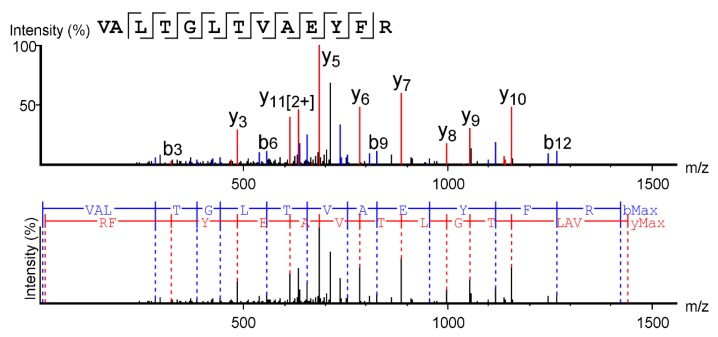
Mass spectrometry (MS/MS) fragmentation was used to obtain the complete sarconesin II aa sequence. The CID spectrum from mass/charge (*m/z*) of its double-charged ion gave [M + 2H]^2+^, *m/z* 720.3984. The ions from y (red) and b (blue) series (marked at the top of the spectrum) represent the primary structure: VALTGLTVAEYFR.

**Figure 4 molecules-24-02077-f004:**
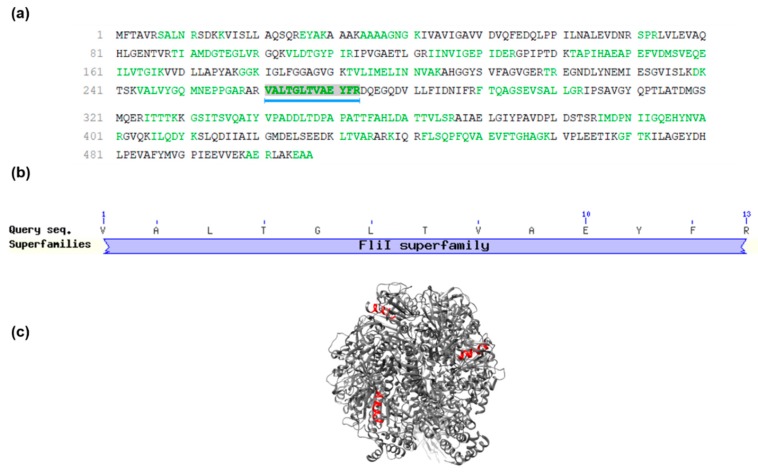
Sarconesin II sequence and conserved domains. (**a**) Sarconesin II spectrum match indicated by a blue line below the mitochondrial KNC23160.1 ATP synthase subunit beta sequence [*Lucilia cuprina*]. Sarconesin II was found between residues 260 and 274, covering 3% of the whole protein sequence. Sequences in green represent tryptic peptides. (**b**) Sarconesin II sequence embedded in *Calliphoridae* multiple sequence alignment search [43] shows the putative conserved domain. Sarconesin II appears as an ATP synthase beta subunit protein conserved residue belonging to the Fli-1 superfamily. (**c**) Representative model of the ATP Synthase Subunit Beta, Mitochondrial (PDB ID: 2w6j) built with Chimera [44]. Sarconesin II is exposed on the surface (red).

**Figure 5 molecules-24-02077-f005:**
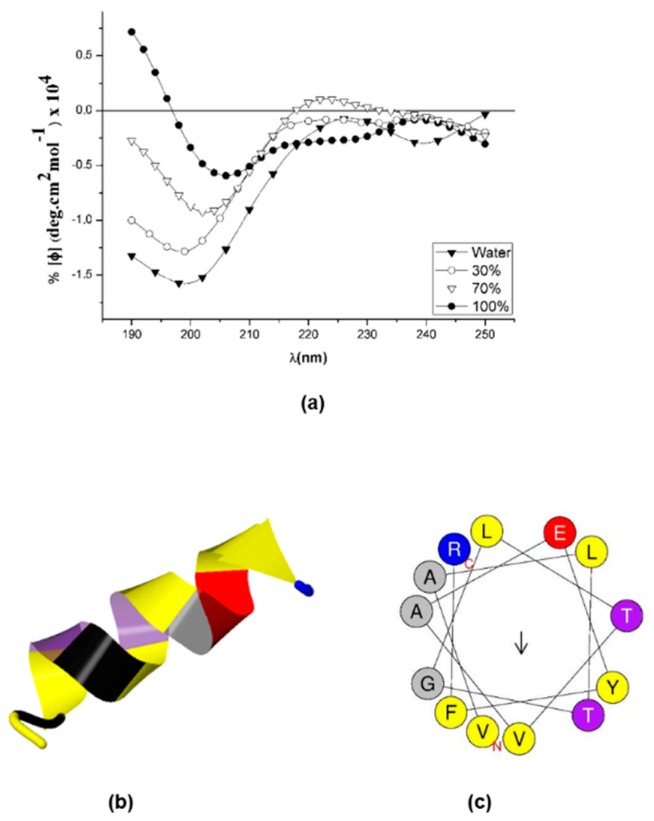
Sarconesin II secondary structure. (**a**) A Jasco-1500 instrument was used for measuring sarconesin II’s circular dichroism spectrum. Sarconesin II CD spectra variation at 0%, 30%, 70% and 100% trifluoroethanol (TFE) concentrations (**b**). I-TASSER sarconesin II secondary structure gave an α-helix, depicted in spiral ribbon format, using common colors. (**c**) Computation of sarconesin II α-helical wheel [54]. No hydrophobic face reported. Note slightly opposite arrangement of hydrophobic (yellow) versus charged (red, blue) aa.

**Figure 6 molecules-24-02077-f006:**
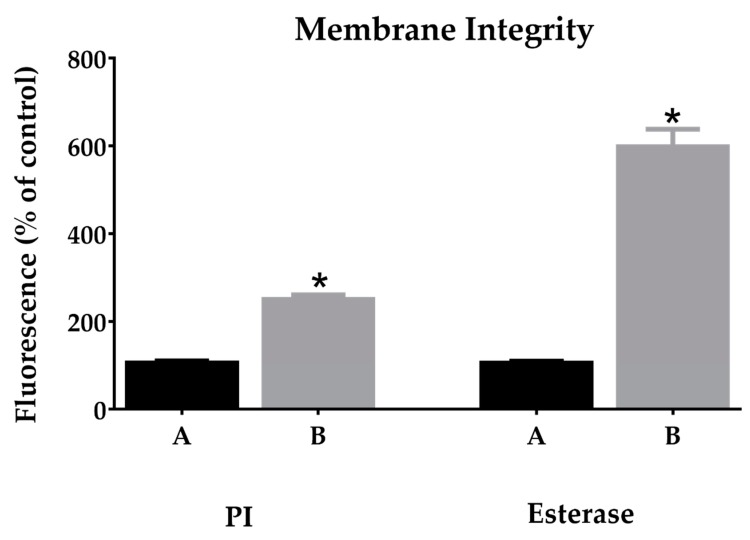
PI and esterase change mean fluorescence of bacteria treated with sarconesin II MIC concentration (B). PI incorporation showing membrane-damaged bacteria. Treated cells (B) showing increased PI and CFDA fluorescence when incubated with sarconesin II. Esterase activity determined by 5,6-carboxyfluorescein diacetate cleavage, expressed as a percentage of control PBS (A) activity. Data is expressed as mean ± SD (*n* = 3). * *p* < 0.001: significantly different from the control.

**Figure 7 molecules-24-02077-f007:**
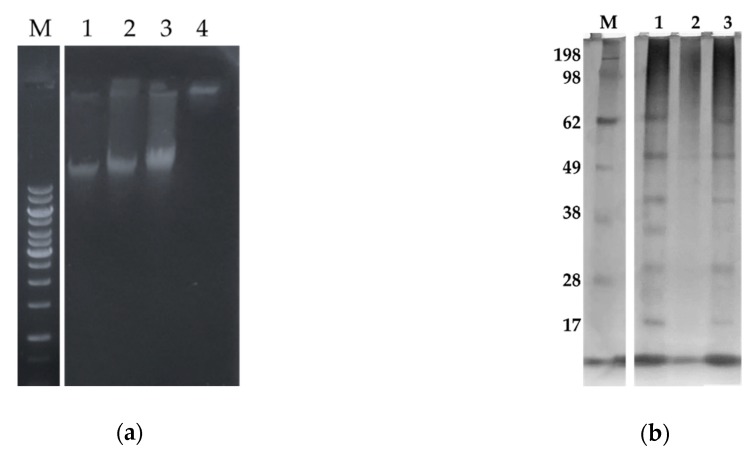
Sarconesin II effects on *E. coli* DNA and proteins assays. (**a**) Gel retardation assay for evaluating sarconesin II effect on DNA. M, GeneRuler 1kb DNA ladder; 1–3, Sarconesin II concentrations were 0, 25, 50 µM, showing proportionally suppressed migration regarding the increased amount of peptide compared to bDNA. The fourth concentration at 100 µM showed that migration was suppressed, suggesting sarconesin II interaction with DNA. (**b**) Changes in protein profile for *E. coli* treated with sarconesin II. Cells treated with or without sarconesin II were washed and sonicated in lysis buffer and the protein profile was analyzed in 12% SDS-PAGE; proteins were visualized by silver nitrate staining. M, molecular weight marker (kDa) (SeeBlue Invitrogen). The streptomycin control (2) revealed no protein profile. There was no difference regarding sarconesin II (3)-treated bacteria compared to the PBS control (1) protein profile, suggesting no peptide action on proteins.

**Figure 8 molecules-24-02077-f008:**
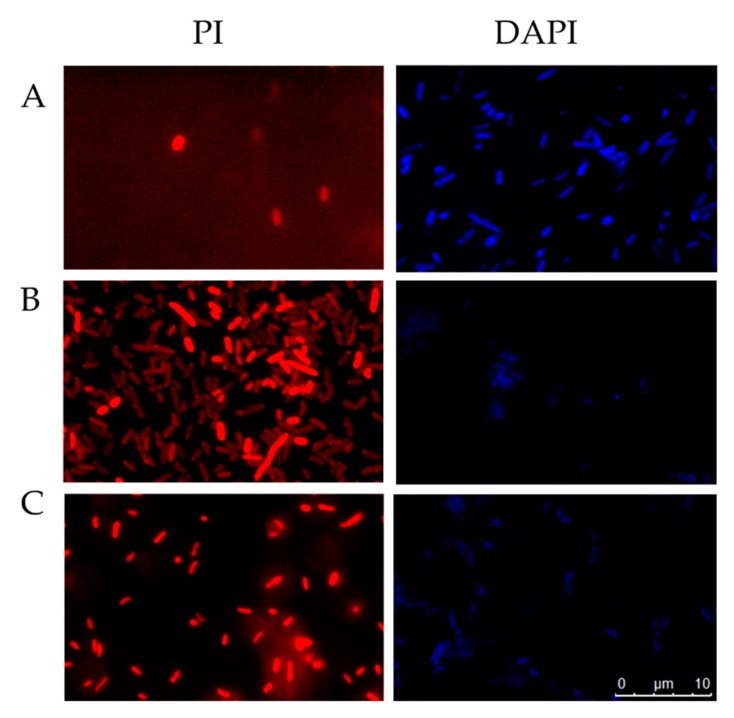
Fluorescence microscopy of *E. coli* cells incubated for 4 h at 37 °C and stained with PI [56] or DAPI (blue). Untreated control cells, PBS (**A**), cells treated with sarconesin II (**B**), cells treated with ampicillin for PI or ciprofloxacin for DAPI (**C**). PI assay revealed bacterial membrane alteration when treated with sarconesin II and DAPI-stained cells had partial fluorescence, showing DNA fragmentation by sarconesin II.

**Figure 9 molecules-24-02077-f009:**
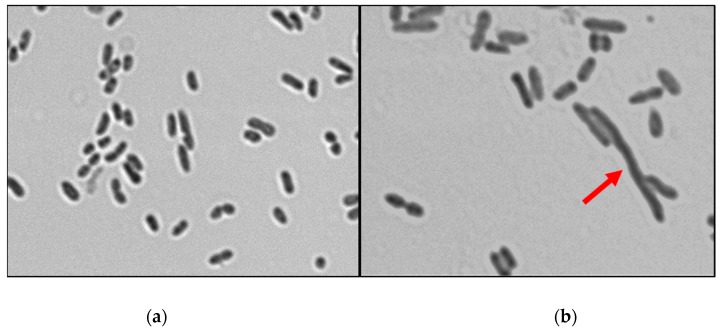
Confocal microscopy image obtained from *E. coli* bacterial cultures incubated for 4 h with PBS (**a**) or sarconesin II (**b**). Cultures were obtained after 4 h incubation at 37 °C. The arrow indicates filamentous cell morphology.

**Figure 10 molecules-24-02077-f010:**
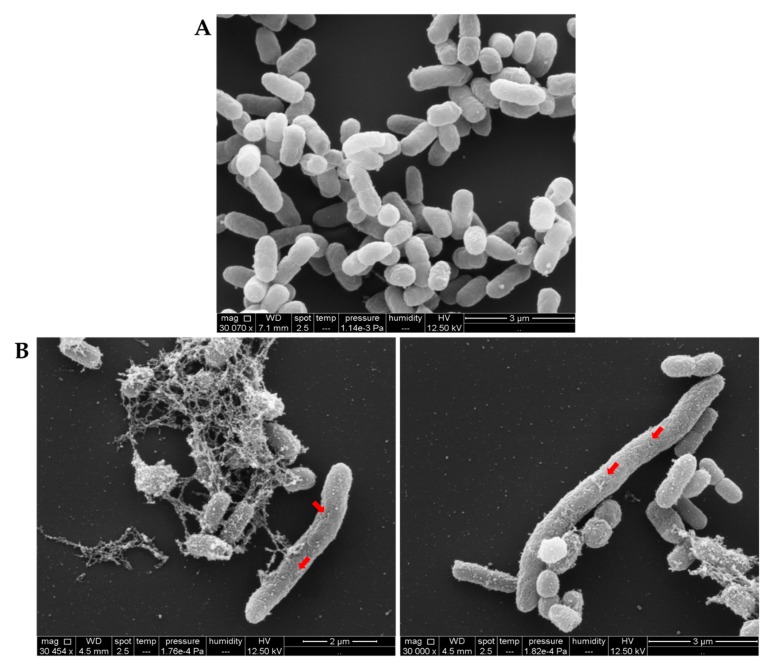
Sarconesin II MIC effect on *E. coli* bacterial membrane by scanning electron microscopy (SEM). Untreated *E. coli* (**A**) had a normal smooth surface, while treatment with sarconesin II (**B**) gave an elongated pattern, membrane disruption and blebbing on the outer face (arrows). Some bacteria had variable length, rough cell surfaces or globular protrusions on their surfaces. These images revealed that sarconesin II could induce alterations in cell morphology.

**Table 1 molecules-24-02077-t001:** Sarconesin II’s antibacterial activity spectrum.

Microorganism	MIC (µM) ^1^
Gram-negative bacteria	
*Escherichia coli* K12 MG1655	7.8
*Escherichia coli* DH5α	3.9
*Pseudomonas aeruginosa* PA14	15.6
*Pseudomonas aeruginosa* ATCC 27853	7.8
Gram-positive bacteria	
*Staphylococcus aureus* ATCC 29213	3.9
*Micrococcus luteus* A270	1.9

^1^ MIC, minimum inhibitory concentration; The MIC refers to the minimal peptide concentration without visible bacterial growth in a liquid medium.

**Table 2 molecules-24-02077-t002:** Sarconesin II’s theoretical physicochemical properties.

Peptide Properties	
Sequence	VALTGLTVAEYFR
Length	13
Molecular weight	1439.67
Formula	C_67_H_106_N_16_O_19_
Theoretical isoelectric point (pI)	5.97
Net charge	0
Molar extinction coefficient (ε)	1490 M-1 cm-1
Instability index	2.70
Aliphatic index	120.00
Grand average of hydropathicity (GRAVY)	0.869

The ProtParam tool in ExPASy was used to obtain physicochemical parameters [41].

**Table 3 molecules-24-02077-t003:** Known antimicrobial peptides having similarity with sarconesin II, as identified in the Antimicrobial Peptide Database (APD2) (Wang et al., 2009).

Peptide Name	Sequence Alignment	Source Organism	APD Identifier	Percentage Similarity
Temporin-HN1 (14 aa)	+ A I L T T L A N W A R K F L V A + L T G L + T V A E Y F R	Frog *Odorrana hainanensis* [46]	AP01959	40%
H4-(86-100) (15 aa)	V V Y A L K R N G R T + + L Y G F + + V + A L + + T G L T V A E Y + F R	Rat [47]	AP02806	38.8%
CcAMP1 (17 aa)	M W I T N G + G V A N W Y F V L A R V A L T + G L T V A + E Y F + + + R	Stink bug *Coridius chinensis* [48]	AP02595	38.88%
Plantaricin DL3 (20 aa)	V G P G A I N A G + T Y L V S R E L F E R V + + + A + L T G L T + + V + A E Y F + R	Probiotic *Lactobacillus plantarum* DL3 [49]	AP02979	38.09%
VmCT1 (13 aa)	+ F L + G A L W N V A K S V F + V A L T G + L + T V A + E Y F R	Scorpion *Vaejovis mexicanus smithi* [50]	AP02216	37.5%

The Antimicrobial Peptide Database (APD) prediction tool was used to align sarconesin II [45].

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
