# Peer review of "Sarconesin II, a New Antimicrobial Peptide Isolated from Sarconesiopsis magellanica Excretions and Secretions"

_molecules, 2019, doi:10.3390/molecules24112077_

Round 1

Reviewer 1 Report

The manuscript has numerous errors in grammar and expression.

Please define excretions and secretions and the differences between them. Why use excretions and secretions together?

Data are presented for only fraction 36 from the 75 fractions obtained.There are no data presented for the other 74 fractions. Some of the fractions other than fraction 36 should show some alboit lower antimicrobial activity than fraction 36.It appears that the data are so clearcut.

From the chromatogram it appears that the sample was a highly heterogeneous mixture. The peak corresponding to  fraction 36 did not go down to somewhere near the baseline. How was fraction 36 collected and how could fraction 36 be purified to the peak which went down all the way to the baseline in a SINGLE chromatographic step? It appears that the data are so clearcut.What is the proof of purity?

There are no data on increase in specific activity with  purification.These should be provided in any study reporting isolation.The inclusion of data on the fold of increase in specific activity especially in a protocol that involved only one or two steps is essential.

 There are no data on MBC which are critical.

Fig 2. Data points are important and needed between time=120 minutes and time=240 minutes because there was an abrupt decline between  time=120 minutes and time=240 minutes.

There was  a statistically significant STIMULATION  of growth of the HeLa CANCER CELLS at 25 uM and 50 uM which was not discussed. Antimicrobial peptides often suppress rather then promote growth of cancer cells.

Fig 2.How was 0.02% hemolysis at 50 uM and 0.05% hemolysis at 100 uM concentration of the antimicrobial peptide determined?0.02% and 0.05% are very small values.

Fig 6A  and 6B. p should be <0.001 and NOT < 0.05.

Table II. What is meant by "theoretical"?Are the parameters not the actual experimental data acquired?

The authors expect the use of isolated antimicrobial peptide to solve the problem of microbial resistance to drugs.They should have tested the antimicrobial peptide on DRUG-RESISTANT strains.

Please explain how the structural resemblance of the isolated antimicrobial peptide to the beta subunit of ATPase synthase can lead to its antimicrobial activity.

In what ways is the isolated antimicrobial peptide superior to the known antimicrobial peptides and proteins?

Author Response

All replies are in the attached Microsoft Word document.

Reviewer 2 Report

In this paper, Díaz-Roa et al. reported the antimicrobial activity of sarconesin II from blowfly larval and aimed to understand its mechanism of action against bacteria. The present manuscript is interesting and this antimicrobial peptide with low toxicity is potent, providing a good candidate for addressing the globally urgent resistant strain problem. The experimental idea is purposeful and provides some important evidence and idea in guiding the research of antimicrobial peptides.

However, there are some concerns need to be addressed before this paper could be accepted for publication. 

1. The concentration of DMSO used as positive control in MTT assay is unknown, the percentage of DMSO should be mentioned. 

2. This paper declared that amino acid sequence of sarconesin II has been determined by mass spectrometry. How to confirm the leucine and isoleucine in the peptide sequence by MS/MS In figure 3? These two amino acids have identical mass. 

3. I doubt that the conclusion of E. coli DNA oxidative damage by sarconesin II is not well supported by the data from DAPI fluorescent staining.

4. Where does the data “The time that sarconesin II remained stable without degradation in E.coli was 10 h” come from? If the authors have performed relevant experiment please provide a result, otherwise, it needs a citation.

5. On page 8, Line 174, please delete the second “using” in “using the using HeLa cell line”

6. There are several abbreviations and corresponding full name appeared in the text. Keep the spell consistent in the text, such as Escherichia coli and E.coli, as well as Pseudomonas aeruginosa and P. aeruginosa in table 1 and the text.

Author Response

All the replies are in the attached Microsoft Word document.

Reviewer 3 Report

Dear Authors,

I read the manuscript “Sarconesin II, a new antimicrobial peptide isolated from Sarconesiopsis magellanica excretions and secretions” very carefully. Although the paper is appropriate for publication in Molecules, I accepted it with major revisions because there are a lot of grammatical errors.

1) Authors should read the paper carefully and, if it is possible, they should use an English editing service.

For example:

Page 2 of 46, line 45, the sentence “Antibiotic resistance is to dangerous levels” should be “Antibiotic resistance is at dangerous levels”;

Page 2 of 46, line 49, the sentence “This study was thus aimed at identifying and characterising” should be “This study aims to identify and characterize”;

Page 3 of 46, line 54-55, the sentence “The molecule in the ES was characterised as sarconesin II and had activity against Gram-negative” should be “The molecule in the ES was characterized as sarconesin II and it showed activity against Gram-negative”.

(A lot of grammatical errors are found throughout the paper. Please, check the text carefully)

2) The first time a pathogen is named, the Authors should write the full name. For example:

Page 5 of 46, line 110-111, the sentence “K. pneumoniae” should be “Klebsiella pneumoniae”.

3) There are many typing errors in the text. For example:

Page 3 of 46, line 71: replace “synthesised” with “synthesized”

Page 5 of 46, line 121: replace “a universal” with “an universal”;

Page 6 of 46, line 132: replace “characterised” with “characterized”;

Page 6 of 46, line 143: replace “lyophilised” with “lyophilized”;

Page 8 of 46, line 174: replace “using the using HeLa cell line” with “in HeLa cell line”;

Page 16 of 46, lines 322 and 336: replace “analysed” with “analyzed”;

Page 16 of 46, line 337: replace “visualised” with “visualized”;

Page 28 of 46, line 607: replace “permeabilisation and depolarisation” with permeabilization and depolarization”;

Page 28 of 46, line 608: replace “destabilisation” with “destabilization”.

(Please, check the text carefully)

4) In Table 1, “E. coli DH5α” should write with the full name.

Figure 1, page 7 of 46, line 150: replace “TFA/H2O” with “TFA/H2O”

5) Page 8 of 46, line 173-174, the sentence “was used for determining” should be “was used to determine”

Page 9 of 46, line 194-195, the sentence “was used for analyzing” should be “was used to analyze”

Page 10 of 46, line 207 and line 222, the sentence “was used for obtaining” should be “was used to obtain”

Page 16 of 46, line 320, the sentence “which is why it did not migrate” should be “which is because it did not migrate”

Page 18 of 46, line 367, the sentence “revealed differing morphologies” should be “revealed different morphologies”

Page 20 of 46, line 401, the sentence “third instar maggot” should be “third-instar larvae”

Page 30 of 46, line 671, the sentence “were used for evaluating” should be “were used to evaluate”

Page 32 of 46, line 708-709, the sentence “were used for classifying” should be “were used to classify”

Page 32 of 46, line 712-713, the sentence “was used for obtaining” should be “was used to obtain”

Page 33 of 46, line 750, the sentence “further treatment at differing sarconesin II concentrations” should be “further treatment at different sarconesin II concentrations”

Author Response

(The authors gave the same response as above.)

Round 2

Reviewer 1 Report

The revised manuscript shows improvement over the previous version.

Reviewer 3 Report

The Authors have corrected all reported errors and also have improved the manuscript, so it can be accepted in present form.

Best regards.